# An Exploratory Critical Review on TNF-α as a Potential Inflammatory Biomarker Responsive to Dietary Intervention with Bioactive Foods and Derived Products

**DOI:** 10.3390/foods11162524

**Published:** 2022-08-21

**Authors:** Stefano Quarta, Marika Massaro, Maria Annunziata Carluccio, Nadia Calabriso, Laura Bravo, Beatriz Sarria, María-Teresa García-Conesa

**Affiliations:** 1Department of Biological and Environmental Sciences and Technologies (DISTEBA), University of Salento, 73100 Lecce, Italy; 2National Research Council (CNR), Institute of Clinical Physiology (IFC), 73100 Lecce, Italy; 3Institute of Food Science, Technology and Nutrition (ICTAN), Spanish National Research Council (CSIC), José Antonio Nováis 10, 28040 Madrid, Spain; 4Research Group on Quality, Safety and Bioactivity of Plant Foods, Centro de Edafología y Biología Aplicada del Segura (CEBAS), Spanish National Research Council (CSIC), Campus de Espinardo, 30100 Murcia, Spain

**Keywords:** bioactive compounds, obesity, inflammation, interindividual variability, (poly)phenols, human clinical trials, genotype, genetic variants

## Abstract

This review collects and critically examines data on the levels of tumour necrosis factor-alpha (TNF-α) in lean, overweight and obese subjects, and the effects of intervention with different foods and food products containing bioactive constituents in overweight/obese individuals. We additionally explore the influence of different single nucleotide polymorphisms (SNPs) on TNF-α levels and compare the response to food products with that to some anti-obesity drugs. Our aim was to provide an overview of the variability, consistency, and magnitude of the reported effects of dietary factors on TNF-α, and to envisage the reliability of measuring changes in the levels of this cytokine as a biomarker responsive to food intervention in association with the reduction in body weight. Regarding the circulating levels of TNF-α, we report: (i) a large intra-group variability, with most coefficients of variation (CV%) values being ≥30% and, in many cases, >100%; (ii) a large between-studies variability, with baseline TNF-α values ranging from <1.0 up to several hundred pg/mL; (iii) highly variable effects of the different dietary approaches with both statistically significant and not significant decreases or increases of the protein, and the absolute effect size varying from <0.1 pg/mL up to ≈50 pg/mL. Within this scenario of variability, it was not possible to discern clear differentiating limits in TNF-α between lean, overweight, and obese individuals or a distinct downregulatory effect on this cytokine by any of the different dietary approaches reviewed, i.e., polyunsaturated fatty acids (PUFAs), Vitamin-D (VitD), mixed (micro)nutrients, (poly)phenols or other phytochemicals. Further, there was not a clear relationship between the TNF-α responses and body weight changes. We found similarities between dietary and pharmacological treatments in terms of variability and limited evidence of the TNF-α response. Different factors that contribute to this variability are discussed and some specific recommendations are proposed to reinforce the need to improve future studies looking at this cytokine as a potential biomarker of response to dietary approaches.

## 1. The Biological Scenario of TNF-α

Tumour necrosis factor-alpha (TNF-α) is a widely investigated multifunctional cytokine that belongs to the TNF/TNFR (tumour necrosis factor receptor) superfamily. It is produced by multiple cells, primarily by circulatory and infiltrating immune cells, such as macrophages, T and B lymphocytes, natural killer cells, and monocytes, as well as by other types of cells including adipocytes, endothelial cells, muscle cells, mast cells, fibroblasts, osteoclasts [1,2] (Figure 1). The soluble trimeric form of TNF-α is activated by binding to the receptors TNFR1 and TNFR2 and triggering multiple cell-signalling transduction pathways involved in many different processes, i.e., cell proliferation and differentiation, cell death (apoptosis) and survival, and cell communication. Maintaining normal physiological levels of TNF-α is critical for health, and TNF-α itself can contribute to this process by being involved in a negative feedback regulatory mechanism between different subpopulations of T cells (effector and regulatory T cells) [3].

TNF-α is also a potent activator of the production of different cytokines promoting a pro-inflammatory status and thus, it has an important function in the inflammatory and immune responses. Multiple molecular and cellular mechanistic studies have pointed to the therapeutic potential of targeting the system TNF/TNFRs for the treatment of inflammatory and immune diseases [4]. In general, the long-term elevated levels of TNF-α associated with diseases such as arthritis, inflammatory bowel disease (IBD) or some specific types of cancer [5,6,7] can be deleterious, and the reduction in the levels of this cytokine is generally considered a beneficial anti-inflammatory effect. Thus, anti-TNF-α therapy has been successfully applied to combat some of these diseases even though it can also have some serious adverse effects [8].

It is now also clear that TNF-α has an impact on healthy metabolism as well as on metabolic diseases, particularly, on obesity-linked glucose metabolism and insulin resistance (IR) [9]. Increased levels of TNF-α have been linked with obesity and the associated low-grade chronic inflammatory status as well as with the derived cardiometabolic disorders [10,11]. Inversely, the reduction in body weight has been associated with a decrease in the levels of this and other cytokines [12]. Intriguingly, TNF-α also relates to metabolic dysfunctions such as anorexia and cachexia which share with obesity overlapping inflammatory mediators and IR but opposite relationship with fat mass [13,14,15]. Further, the use of anti-TNF-α antibodies (adalimumab, certolizumab, infliximab, etanercept) against some inflammatory diseases (e.g., IBD, Crohn’s disease, rheumatoid arthritis, psoriatic arthritis, spondyloarthritis) has produced contrasting results with some studies reporting increases in body mass index (BMI) and other anthropometric parameters upon treatment [16,17,18]. Overall, the mechanisms explaining the relationship between TNF-α, body weight changes, and metabolic disorders are not yet clear. It remains thus essential to clearly understand and establish the relevance of the levels of TNF-α and of its modulation (size and direction of its fluctuation) in the context of lean physiology and increased body weight (overweight/obese) conditions. Different sets of immune cells infiltrated into metabolic organs such as the adipose or the liver tissues regulating, at least partly, the balance between metabolic homeostasis and dysfunction (Figure 1). In obesity, the accumulation of fat and enlargement of adipocytes contribute to the dysfunctionality of these cells as well as that of endothelial cells and macrophages (M1-polarized macrophages), increasing the secretion of a range of adipokines including TNF-α [2]. These elevated levels of TNF-α have been associated with impaired insulin signalling [19], and altered lipid metabolism (lipolysis/lipogenesis) [1]. It has also been indicated that TNF-α may lead to weight loss by interfering with central weight regulation in the brain through the stimulation of anorexigenic neuropeptides release as well as by influencing catabolic processes in peripheral muscle cells [17].

Overall, TNF-α is a pleiotropic molecule involved in multiple physiological and pathological mechanisms and constitutes an important target in therapies against a diversity of immunologically related disorders. In the context of obesity, a broad range of human intervention dietary studies have investigated the changes on the blood levels of TNF-α in connection with body weight changes. In this manuscript, we present an exploratory and critical revision of those studies looking at the modulation and variability in the levels of this cytokine. We examine a selection of human intervention trials with different dietary approaches including foods and food derived products containing (micro)nutrients and/or bioactive constituents. The number of studies selected and included in each section of this review is indicated in Figure 2. Also, we critically investigate the reported differences in the levels of TNF-α between lean, overweight and obese individuals, as well as the influence of specific genetic variants to this variability. Additionally, we compare the nutritional intervention results to those attained with specific anti-obesity drugs. Our main goals were to contribute to the understanding of the reliability of measuring the levels and changes in this cytokine, and to reinforce the need to improve future trials so that TNF-α may be used as a reliable biomarker of chronic inflammation associated with obesity in response to dietary intervention.

## 2. TNF-α Levels in Lean, Overweight and Obese Individuals

A good diagnostic biomarker has been defined as one that allows for the differentiation between the conditions under investigation (health and disease), and for identifying an individual within one of those conditions [20]. When reporting the responses of a biomarker to any treatment, i.e., dietary or drug intervention, it is thus essential to try to describe and understand the relevance of the response (both size and direction of change) within the context of the interindividual variability but also taking into consideration the differences in the values between the investigated conditions.

We reviewed a number of human studies carried out between 1995 and 2022, where the circulatory levels of TNF-α were determined in lean, overweight and obese individuals [21,22,23,24,25,26,27,28,29,30,31,32,33,34,35,36,37,38,39,40,41,42,43,44,45]. The results of all the studies evaluated in this section and the differences between obese and lean individuals for each study are specified in the Appendix A. We examined and compared all these data in an attempt to visualize the difference in the levels of TNF-α between these metabolic conditions, and to assess whether this difference could be used with sufficient reliability to clearly differentiate between lean, overweight and obese individuals. The levels of TNF-α were most commonly reported in pg/mL and as the mean values ± standard deviation (SD) for lean, overweight, and obese individuals. We estimated the coefficient of variation (CV%) (using the standard formula as follows: (CV%) = (Standard Deviation/Mean) × 100) for each of the different subgroups in every study. A summary of the intragroup variability and levels of circulatory TNF-α in lean, overweight, obese, and very obese individuals as reported in the different human studies examined in this review is displayed in Table 1.

The results show that the CV% values were typically >30% and, in some cases, well above 100%, for the lean, overweight and obese subpopulations, indicating a rather high relative intragroup variation. The data also show that the circulating levels of TNF-α displayed a very high between-studies variability. In the lean individuals, reported values expanded from <1.0 pg/mL to ~80 pg/mL; however, in most cases, the values were found in the range 2.0–6.0 pg/mL (Table 1). In general, most of the investigated studies reported higher levels of TNF-α in the overweight, obese, and/or very obese subgroups than in the lean participants. However, the differences were (S) in some studies [23,24,25,26,27,30,33,34,35,36,37,38,40,42] and not significant (NS) in others [21,22,28,29,31,38,39,44,45]. Also, the magnitude of the difference was highly variable. For example, in Polish women, the levels of TNF-α varied significantly (*p* < 0.001) from 2.9 ± 2.2 pg/mL in the lean participants to 6.5 ± 3.1 pg/mL in overweight individuals, 6.8 ± 3.1 pg/mL in participants with obesity, and 7.4 ± 2.6 pg/mL in very obese participants [23]. In another study also conducted in Polish women, the differences reported between lean and obese participants were 4.5 ± 2.3 pg/mL and 8.8 ± 7.0 pg/mL, respectively (*p* < 0.05) [30]. In a sample population from Israel [24], the difference found between obese and non-obese participants was also very significant, but the levels of the cytokine were much smaller (1.0 ± 0.8 pg/mL vs. 0.3 ± 0.3, *p* < 0.001). As an example of (NS) but substantial differences between individuals, in a sample population of Spanish women, the levels of TNF-α ranged from 5.7 ± 9.8 pg/mL in lean participants to 11.6 ± 19.0 pg/mL in obese ones, and 8.6 ± 12.4 pg/mL in very obese participants [22]. In addition, a few studies also reported lower levels in the overweight/obese participants than in the lean ones [32,41,43]. Likewise, a study conducted in a sample mixed population in the USA indicated that the levels of TNF-α were 0.8 ± 7.2 pg/mL in the obese participants and 82.3 ± 89 pg/mL in the lean ones [32]. Further results can be found in Appendix A.

Overall, the percentage of change from lean to overweight or obese subgroups varied from a (NS) 2% increase [41] up to a maximum increase of ~1700% in some morbidly obese individuals [37], although most increases oscillated between ~20% and ~200%. The levels of TNF-α in the overweight subgroups ranged from <1.0 pg/mL up to 30 pg/mL, and in obese participants, from <1.0 pg/mL up to 294 pg/mL. In these two groups, however, most values oscillated between ≈3.0 and 6.5 pg/mL in overweight individuals and between ≈1.0 and 10 pg/mL in the obese subjects (Table 1). Participants categorized as very obese also exhibited highly variable levels of TNF-α ranging from ~1.0 pg/mL up to values as high as 713 pg/mL (Table 1). Further, the combination of being overweight or obesity with an additional disorder such as Type 2 Diabetes mellitus (T2D), Metabolic Syndrome (MetS) or Polycystic Ovary Syndrome (PCOS) did not appear to critically modify the levels of this cytokine with values also varying between <1.0 pg/mL up to 20 pg/mL (Appendix A).

The overall scenario regarding the circulatory levels of TNF-α in humans shows a large intra- and between studies variability with several hundred- and, even up to several thousand-fold differences within the values attributed to each specific group of individuals. Many of the studies included in this section reported an increase in the levels of this cytokine in the overweight/obese sample populations as compared to the lean ones, and the general message conveyed is that there is an association between increased body weight and increased levels of the cytokine. Yet, looking at all the data gathered here, it was not possible to designate clear-cut ranges of TNF-α values that differentiated between lean, overweight and obese individuals, and/or to discern any clear tendency between TNF-α levels and body weight. The majority of the reported values for circulating levels of TNF-α ranged between ~1.0 and 10.0 pg/mL, with apparent independence of the body weight category posing some reasonable doubts on the potentiality of TNF-α as a biomarker to differentiate between body weight conditions.

## 3. Human Intervention Studies Looking at the Effects of Foods and Derived Products on the Levels of TNF-α

The most common, accessible, and first-line options for the treatment of body weight excess are behavioural changes which consist principally of adopting a healthier lifestyle, including a change in eating habits towards a healthier diet (e.g., low-calorie diet (LCD), Mediterranean diet (MD)) [46,47]. General recommendations against obesity include the specific reduction in the consumption of saturated fats and carbohydrates as well as the increased intake of plant foods. Many epidemiological and experimental studies have shown that the consumption of colourful fruits and green vegetables, which are rich in a natural blend of bioactive constituents, i.e., vitamins, fibre, phytochemicals, can improve the metabolic profile and reduce the risk of developing chronic diseases [48]. Therefore, consumption of fruits and vegetables has become a cornerstone of most dietary recommendations aimed at healthy body weight management [49]. Additionally, supplementation of the diet with a variety of those bioactive constituents including specific (micro)nutrients such as polyunsaturated fatty acids (PUFAs), vitamins, minerals, amino acids, fibre, and (or) phytochemicals ((poly)phenols, carotenoids, terpenoids, etc.) has also been widely investigated for potential metabolic and body weight benefits. These bioactive compounds have been mostly administered as (semi)purified mixed extracts, or as the constituents of enriched foods and beverages. In the following sub-sections, we have tabulated, described, and critically examined the results of a series of intervention studies (mostly randomized clinical trials (RCTs) with parallel or crossover designs) set up to investigate the effects of different foods and derived products to modulate body weight, with a focus on the TNF-α changes observed in response to dietary treatment. For each study and where it was not reported, we have calculated the intra-group CV% and the effect size (described as the net difference between the effect in the treatment (T) and control (C) groups).

### 3.1. Dietary Interventions with PUFAs

Dietary fatty acids (FAs) constitute essential nutrients and an important source of energy, having a considerable range of biological functions including a modulatory role of metabolism and immune responses and inflammation [50]. Depending on their structure and degree of saturation, the different FAs can influence various aspects of human metabolic physiology, such as affecting the feeling of satiety [51] or regulating blood lipids and inflammation [52]. While saturated FAs have been considered detrimental to health, monounsaturated (MUFAs) and PUFAs, of which fatty fish are particularly rich, appear to offer potential health benefits [53]. In particular, ω-3 PUFAs, such as the eicosapentaenoic acid (EPA) and the docosahexaenoic acid (DHA), have been reported to have cardiometabolic and anti-inflammatory properties [50,54]. These (micro)nutrients could help in the treatment and prevention of body weight excess and associated inflammatory conditions and disorders. Cell culture and animal models of obesity [55] have postulated a range of potential mechanisms by which ω-3 PUFAs may exert anti-obesogenic effects, such as the promotion of mitochondrial biogenesis and fatty acid-β oxidation [56], the inhibition of adipocyte differentiation, and (or) the induction of apoptosis in pre-adipocytes [57]. Importantly, these FAs have also been shown to reduce the secretion of inflammatory cytokines, including Monocyte Chemoattractant Protein-1 (MCP-1), Interleukin-6 (IL-6), and TNF-α, in human adipose tissue as well as in mature adipocytes challenged with pro-inflammatory molecules [58]. These results have motivated the study of the effects of the consumption of ω-3 PUFAs on human body weight and body composition as part of a weight-loss strategy [59]. Nevertheless, the current evidence in adults [60,61] and adolescents [62] remains weak and contradictory.

In this section we reviewed a series of intervention trials in which overweight/obese participants received supplements containing ω-3 PUFAs, and the changes in body weight and TNF-α levels were investigated [63,64,65,66,67,68,69,70,71] (Appendix A). These PUFAs were administered in the form of encapsulated oil to adults of both sexes [63,66,67], except for two trials that were performed on teenagers [64,65]. The intervention period lasted from ≈30 d to 180 d. The placebo (PLA) group consisted of vegetable oils such as sunflower oil, corn oil (both rich in linoleic acid), soybean oil [63,66], medium-chain triglycerides [64] or starch [65]. No other lifestyle modifications were reported, except for one study in which the subjects followed a very LCD (in both the control and experimental groups) [67]. The administered doses of ω-3 PUFAs ranged from ≈500 to 4000 mg/d, and the circulatory levels of TNF-α were mostly determined by measuring protein concentration in serum or plasma using an ELISA assay [63,64,65,66,67,68,69,70]. All these studies displayed a considerable within-group variability with some studies reaching CV% values well above 30%. Baseline levels of circulating TNF-α varied from ≈1.0 to 26.0 pg/mL. Following exposure to the ω-3 PUFAs, the modulation of those levels (calculated as the difference between the changes observed in the T and C groups, i.e., before and after intervention) ranged from a small (NS) reduction in −0.02 pg/mL [63] to an (NS) increase of +10.9 pg/mL [66]. Two studies reported significant reductions in TNF-α. In the study conducted by Lopez-Alarcon et al. [65] this reduction was not supported by numerical data, whereas only Dangardt et al. [64] reported a very significant (*p* = 0.008) reduction of −0.50 pg/mL. Regarding body weight changes, none of the studies showed significant (S) effects from the supplementation with ω-3 PUFAs. In the study by de Luis et al. [67] that tested the effects of the combination of DHA with a LCD, a significant reduction in body weight was observed both in the experimental and control groups, with no significant difference from the supplementation with DHA.

The results of the supplementation with several foods or oils enriched in PUFAs [68,69,70,71] are also presented in Appendix A. The nutritious seeds of chia (*Salvia hispanica* L.) have been attributed metabolic and immune-regulatory properties partly due to their high content of PUFAs [72]. An intervention trial conducted on overweight/obese participants daily supplemented with 50 g of chia seeds showed, however, (NS) effects on body weight and TNF-α levels (small increase of +0.11 pg/mL both in men and in women) after 84 days of intervention [68]. Similarly, daily intervention with 40 g of flaxseeds, another important vegetal source of PUFAs and, in particular, of ω-3 PUFAs [72,73], did not show any significant alteration in body weight or in the levels of the cytokine (−0.2 pg/mL) in obese participants ([69]. Black cumin (*Nigella sativa* L.) oil, also a rich source of PUFAs [74], was tested in Iranian obese women (3 g/d, 56 d) with no effects on body weight. In this study, however, the authors reported a (S) reduction in the levels of TNF-α (−6.3 pg/mL, *p* = 0.03) [70]. Baru almonds (*Dipteryx alata* Vog., legume seeds) also constitute a rich source of PUFAs [75], and were investigated for their regulatory metabolic and anti-inflammatory effects in Brazilian overweight/obese women. Following 56 days of supplementation with 20 g of nuts per day, (NS) changes in body weight were reported and TNF-α was only slightly and not significantly increased (+0.7 pg/mL) [71].

### 3.2. Vitamin D

Vitamin-D (VitD) occurs naturally as cholecalciferol or vitamin D3, and ergosterol or vitamin D2. In humans, the quantitatively most important source of vitamin D3 is that synthesized in the skin by the sun’s ultraviolet rays. Natural dietary sources of vitamin D2 are some fungi (sun-dried mushrooms) and yeast, whereas vitamin D3 can be found in salmon, fatty fish, fish liver oil, organs (e.g., liver, kidneys), meat and egg yolks. Since only few foods contain high levels of VitD, fortification of different food products such as margarine, milk and cereals with VitD is common in many countries. Furthermore, the use of oral supplements is widely recommended in order to prevent deficiencies. The recommended intake of VitD is 10 µg/d (400 IU/d). [76]. VitD plays an essential role in calcium and phosphorus homeostasis and bone health. Since this vitamin binds the vitamin D receptor (VDR), thereby triggering an effect on gene transcription and protein synthesis in many different cells and tissues, this hormone also functions as a wide-range regulator of many cellular homeostasis-, immunity-, and metabolism-related processes. For these reasons, VitD has also been broadly investigated for its role in diseases like cancer, cardiovascular disease (CVD), diabetes and inflammatory diseases [76,77]. Overall, the deficiency of VitD has been associated with some of those diseases but supplementation of the vitamin is not yet clearly related to a beneficial outcome [76]. VitD is largely stored in the adipose tissue, where this vitamin exerts an important regulatory role in the adipocyte’s physiology [78]. Consistent with this, in vitro studies in human and mouse adipocyte cell cultures have shown that the VitD regulates adipose tissue differentiation and growth through multiple mechanisms, including inhibition of pre-adipocyte differentiation, inhibition of FA synthesis, reduction in lipid accumulation in vacuoles, and induction of pre-adipocyte apoptosis [79,80,81]. With regard to the pro-inflammatory phenotype developed by hypertrophic and hyperplastic adipose tissue, studies on human differentiated adipocytes have shown that VitD is also able to inhibit the pro-inflammatory signalling pathways activated by TNF-α and lipopolysaccharide (LPS) [82,83,84,85].

In humans, low plasma concentrations of VitD (hypovitaminosis D) have been consistently reported in overweight/obese people [86], and thus the effects of the supplementation with this vitamin on body weight have been repeatedly investigated. A recent meta-analysis examining the benefits of VitD on various measures of adiposity in healthy overweight and obese adults indicated a lack of effect on BMI, waist circumference (WC), and waist-to-hip ratio. Regarding the pro-inflammatory phenotype associated with overweight/obese individuals, the results of the supplementation with VitD were also unclear [87]. While BMI has been causally associated with several inflammatory markers (including TNF-α), mediation analyses performed to establish causality between serum VitD and serum inflammatory markers indicated no role for this vitamin as a causal mediator between BMI and systemic inflammation. The evidence of the beneficial effects of VitD as a therapeutic agent to improve inflammation in overweight and obese subjects remains poor [88,89].

We revised the results of several intervention trials in which overweight/obese adults of both sexes were treated with VitD, and changes both in body weight and levels of TNF-α were investigated [90,91,92,93,94] (Appendix A). The intervention period varied from 56 to 365 d and the comparison was made against a PLA group which mostly consisted of vegetable oils, e.g., soy oil or a VitD–free oil (Migliol oil) [90,92], or microcrystalline cellulose [91], with no other lifestyle changes. The dose of VitD administered was between 3000 and 4000 IU/d (0.075–0.1 mg/d). In one study [94], the VitD was first given as a unique bolus of 100,000 IU/d (2.5 mg/d) followed by 4000 IU/d. TNF-α was determined mostly by measuring protein concentration in serum using an ELISA assay with baseline average levels ranging from 1.5 to 29.6 pg/mL. The results of these studies showed high within-group variability, with most CV values in the range of >30% to >100%. Reported changes in TNF-α levels following intervention were (NS) increases or decreases with values ranging from −3.5 pg/mL to +0.8 pg/mL, and only one study declared a (S) but rather small reduction in the TNF-α levels (−0.58 pg/mL, *p* < 0.05) [90]. Consistent with previous studies [87], the trials reviewed here corroborated a lack of significant effects of VitD on body weight. Overall, the data analysed in this section show no indication of the role of VitD as an anti-inflammatory agent and in controlling body weight in overweight/obese individuals.

### 3.3. Mixed (Micro)nutrients

A series of trials that investigated the effects on body weight and TNF-α in overweight/obese participants following intervention with other bioactive (micro)nutrients have been examined [95,96,97,98,99,100] (Appendix A). The studies were primarily conducted in mixed populations of men adult men and women, with duration periods between 28 d and 90 d. The levels of protein were measured using an ELISA assay in serum [96,97,98,99,100], except for one study where the levels of TNF-α were measured in plasma [95]. The CV% values ranged from <10% in some cases up to 40–50% in several groups. The data regarding the modulation of the levels of TNF-α were variable and the effect of the different treatments in comparison with the PLA yielded only (NS) results ranging from a reduction of −56 pg/mL with the mixture of leucine and vitamin B6 (NuFit) [95] to an increase of +0.61 pg/mL with the black soy peptide [97]. Intervention with a NuFit active blend (leucine and vitamin B6) [95], L-arginine [96], zinc [98], and hydrolysed cod protein [100] did not show any reducing effects on body weight. Whereas, body weight was significantly reduced following the intake of black soy peptide [97], and WC was also reduced with the intake of yeast β-glucan [99].

### 3.4. Phytochemicals

Phytochemicals cluster a large group of organic non-essential secondary metabolites with rich structural diversity produced principally, but not exclusively, by plants. (Poly)phenols constitute one of the largest and more heterogeneous families of these naturally occurring organic compounds and are characterized by a reactive composition of aromatic rings and one or several phenolic hydroxyl groups [101]. Depending on their chemical structure, (poly)phenols are classified as flavonoids (flavones, flavonols, flavanols, flavanones, isoflavones, anthocyanins) and non-flavonoids (hydrolysable tannins, lignans, stilbenes, phenolic acids) [102]. The main dietary sources of bioactive phytochemicals, and, in particular, of (poly)phenols, are fruits (e.g., apples, grapes, pomegranates, berries, plums, etc.), vegetables (tomatoes, olives, onions, garlic, peppers, cabbage, carrots, etc.), legumes (black beans, black soybeans, etc.), cereals (black rice, rye, black sorghum, purple barley, red sorghum, etc.), chocolate, extra virgin olive oil, and beverages like green tea, black tea, red wine, and coffee [103,104].

A variety of polyphenol-containing products (extracts, freeze-dried powders, foods) have been investigated for a wide range of potential health benefits, including body weight and inflammation regulatory properties. Cell and animal models of obesity have provided extensive supportive evidence of the potential mechanisms by which many of these compounds may exert their activity against obesity, including the inhibition of adipocyte differentiation and proliferation, reduced fat absorption, increased energy expenditure, and increased fat utilization [105]. In addition, an increasing number of intervention studies and various meta-analyses have built up the evidence of the metabolic benefits of the intake of these compounds in humans. In particular, the consumption of ellagitannins-, anthocyanins-, and flavanol-containing tea, cocoa, and apple products have been significantly associated with the reduction in BMI and WC [106,107]. In a more recent meta-analysis, it was reaffirmed that supplementation with anthocyanins (300 mg/d, 28 d) was associated with the reduction in BMI, however, body weight and WC remained unaffected [108]. Regarding the anti-inflammatory properties attributed to the (poly)phenols, there is also much pre-clinical evidence of the downregulation by many of these compounds of both the TNF-α gene expression and protein levels [109,110]. In humans, a meta-analysis examining the effects of anthocyanins on various plasma lipids and inflammatory markers highlighted the ability of these (poly)phenols to selectively modulate TNF-α levels [111]. In a more recent systematic review of human studies looking at the regulatory effects of polyphenol-containing products in post-menopausal women, it was shown that the reduction in the levels of TNF-α was, however, highly variable and inconsistent [112]. It is now well established that the large interindividual variability in response to dietary (poly)phenols poses a substantial degree of uncertainty regarding the consistency and magnitude of the metabolic and inflammatory effects that these compounds may exert in humans. The factors influencing this variability are under thorough investigation, including the complex collaboration between (poly)phenols and gut microflora. A double interaction between them, i.e., the effects of the gut bacteria on the metabolism of (poly)phenols, and the effects of (poly)phenols and resulting metabolites on the gut microbioma composition, are being explored as a major influence on the regulation of metabolic and inflammatory processes and biomarkers, including TNF-α [113].

In this section, we examined the changes in the levels of TNF-α in overweight and obese adults of both sexes following supplementation with different phytochemical-containing products from different sources [114,115,116,117,118,119,120,121,122,123,124,125,126,127,128,129,130,131] (Appendix A). The first part of this table groups a series of studies where the tested products were mixed extracts or powders rich in different (poly)phenols [114,115,116,117,118,119,120,121,122,123,124,125,126,127]. The intervention period varied from ≈20 d up to 360 d, and the comparison was usually made against a PLA group with no other lifestyle modifications, with the exception of one study in which subjects in the control and experimental groups also followed a hypocaloric diet [123]. Regarding TNF-α, the circulating levels of the protein were principally analysed in serum or plasma using an ELISA test. The baseline values were highly variable and oscillated from ≈0.1 [124] to 43.01 pg/mL [127]. Once more, the results of these trials showed high intragroup variability with most CV% values above 30%, reaching values >100% in many cases. Some of the studies reported small and not statistically significant changes in the circulating levels of TNF-α (ranging between −2.6 and +2.6 pg/mL) following treatment with quercetin [114], curcuminoids [115], cocoa extract [117], grape extract (with or without resveratrol) [118,119], grape powder [120,121], and black soybean testa extract [127]. On the other hand, the levels of TNF-α were reported to be significantly decreased following intervention with curcumin (−3.5 pg/mL) or nano-curcumin (−4.8 pg/mL) [116], grape pomace extract (−1.37 pg/mL) [122], grape seed extract (−11.9 pg/mL) [123], pomegranate peel extract (−0.05 pg/mL) [124], freeze-dried strawberries (−2.7 pg/mL) [125], and frozen red raspberries (−11.0 pg/mL *post-pandrial*, −2.1 pg/mL, 28 d) [126]. Most of the studies investigated here did not show or report a (S) reduction in body weight after treatment with the exception of the study by Parandoosh et al. [123] that reported a significant higher reduction in body weight, BMI, WC, and waist-to-hip ratio in the grape seed extract group (300 mg/d containing 85% (poly)phenols) compared to the PLA group. This reduction was not confirmed in any of the other studies conducted with a polyphenol-containing grape product [118,119,120,121,122].

Other mixed products (extracts containing phytochemicals) have also been tested in overweight/obese individuals (Appendix A), i.e., 6 capsules/d of Fruit + Vegetables concentrate powder (containing β-carotene, α-tocopherol, vitamin C, folate, (poly)phenols) [128], 6 capsules/d of Juice Plus+ Premium (2.91 mg carotene, 18.7 mg Vitamin E, 159 mg Vitamin C, 318 μg folate, 6.1 mg lutein, 1 mg lycopene and 0.15 mg astaxanthin) [129], 6 mg/d melatonin [130] and garlic tablets (1000 mg/d containing 2.5 mg of allicin) [131]. Melatonin had a (S) down-regulatory effect on the circulating levels of TNF-α protein (−0.98 pg/mL, *p* = 0.02) [130]. Also, the intake of the Fruit + Vegetable concentrate powder was reported to cause a significant but rather small reduction in TNF-α (−0.07 pg/mL, *p* = 0.035) but only in a small subgroup of participants with C-reactive protein (CRP) values ≥3.0 mg/mL [128]. None of the studies were associated with a body weight-reducing effect.

In addition to those studies conducted with phytochemical-containing capsules, extracts or powders, we have collected some examples of dietary intervention with different foods rich in mixed bioactive (poly)phenols and other phytochemicals which have also investigated for their effects on body weight and TNF-α levels [132,133,134,135,136] (Appendix A). Interventions were carried out with the pulp from the Brazilian juçara fruit (rich in phenolics and flavonoids) [132], Queen Garnet plums (rich in anthocyanins and quercetin) [133], tart cherry juice (rich in anthocyanins, flavonoids and phenolics) [134], tomato juice (mixed carotenoids, phytosterols, phenolics, flavonoids, anthocyanins) [135], and red sorghum biscuit (mixed carotenoids, phytosterols, phenolics, flavonoids) [136]. All the studies were also conducted in obese/overweight adults of both sexes [132,133,134,135,136]. The intervention period varied from 20 to 84 d and the comparison was always against a PLA with no other lifestyle modifications, except for the study conducted with red sorghum biscuits where the participants from both T and C group were on a LCD [136]. TNF-α protein levels were analysed in serum or plasma [133,134,135] or in the supernatant of peripheral blood mononuclear cells (PBMCs) [132] using an ELISA assay. The results of these studies showed high intragroup variability, with CV% values up to 100%. The consumption of tart cherry juice or the sorghum biscuits was associated with similar and rather small (NS) reductions of circulating TNF-α (−0.21 pg/mL and −0.3 pg/mL, respectively) [134,136]. On the other hand, following intervention with the Queen Garnet plums, there was a small but (S) reduction in the levels of TNF-α protein (−2.0 pg/mL, *p* < 0.05) [133]. Supplementation with tomato juice also significantly reduced TNF-α in the entire sample population of overweight/obese participants (−8.3 pg/mL, *p* < 0.05) and in the subgroup of only overweight individuals (−5.3 pg/mL, *p* < 0.05) whereas in the obese participants a large but (NS) increase of TNF-α levels was observed (+21.3 pg/mL) [135]. None of the studies reported a (S) reduction in body weight.

## 4. The Influence of Genetic Variants on TNF-α Levels

As already seen in the previous sections, the circulatory levels of TNF-α show a large variability which may be partly due to a substantial genetic contribution to TNF-α regulation. Nonetheless, the consequences of the presence of different genetic variants on the transcription and final levels of TNF-α are complex and not yet understood. Various single nucleotide polymorphisms (SNPs) have been described within the promoter region of the *TNF-α* gene at different positions (e.g., −308 G/A, −1031 T/C, −863 C/A, −857 C/T, −575 G/A, −376 G/A, −244 G/A, −238 G/A) [137]. One of the most investigated polymorphisms is the −308 G/A (rs1800629) which has been associated with higher serum concentrations of TNF-α as well as with increased susceptibility for obesity, cardio-metabolic disorders (IR, T2D, MetS) and some inflammatory and immune-related disorders [138]. Nevertheless, the results are still rather inconsistent. We have reviewed a number of studies reporting the levels of TNF-α in different populations of mixed adults across the different genotypes for some of those variants, predominantly for the −308 G/A [41,42,137,139,140,141,142,143,144] (Appendix A). As repeatedly indicated in the previous sections, we found a very high intra-subgroup (genotype) variability with most CV% values well above 30% and in many cases >100%. The levels of TNF-α in the homozygous reference subgroups were also very variable with values oscillating between some very low levels, e.g., 0.095 pg/mL, up to values as high as ≈60 pg/mL corroborating large differences between studies. Overall, the differences across the genotypes were rather small, highly variable and mostly (NS) (values ranging from a minimum reduction of −6.0 pg/mL up to a maximum increase of +43.0 pg/mL although most data ranged between −0.8 and +0.5 pg/mL). We only found two studies that reported significant increased levels of TNF-α associated with the presence of the rare variant, in overweight MetS patients (*p* = 0.001, +38.0 pg/mL) [42], and in patients with abdominal aortic aneurism (*p* = 0.045, +50.1 pg/mL) [144]. Our revision suggests that the presence of the −308 G/A does not appear to have a general increasing effect on the levels of TNF-α.

We additionally reviewed a number of studies reporting the levels of TNF-α across the different genotypes for other variants associated with different genes related with obesity and metabolic disorders [137,145,146,147,148,149,150,151,152,153,154,155,156,157,158] (Appendix A). We also found a very high genotype intra-subgroup variability (CV% >40% in most cases), highly variable levels (0.11–90 pg/mL) in the homozygous reference subgroups, and small and mostly (NS) differences across the genotypes except for some specific SNPs. Like so, the TNF-α levels were (S) reduced in Chinese adults with T2D with the rare variant −87 T > C of *PPARD* (*p* < 0.05, −1.7 and −3.1 pg/mL) [146], in malnourished elderly people with the rare variant of rs769214 of *CAT* (*p* < 0.001, −3.2 and −3.9 pg/mL) [153], in patients with heart failure and with the rare genotype for rs1800471 of *TGFB* (*p* = 0.042, −7.0 pg/mL) [154], and in healthy individuals with the rare rs2234632 variant of the gene *ZIP2* (*p* = 0.011, −6.1 pg/mL) [155]. On the other hand, a few other SNPs were found to have more elevated levels of TNF-α in individuals with the rare variant, e.g. FOK1/*VDR* in obese participants (*p* = 0.01 pg/mL, +7.5 pg/mL) [147], and rs141764639/*TMEM182* in normal and obese participants (*p* < 0.05, +4.5 pg/mL and *p* < 0.001, +13.5 pg/mL, respectively) [157]. These results illustrate the complex, multiple and variable increasing and decreasing effects of different SNPs on the circulating levels of TNF-α.

## 5. The Regulatory Effects of Anti-Obesity Drugs on TNF-α

Throughout the years, a considerable number of drugs has been developed and tested to try to combat obesity. Overall, in clinical trials, many of those anti-obesity drugs promote moderate body weight reductions (between ≈ 2.0 to 10.0% after several months of intervention) when compared with lifestyle modifications. Often, those drugs also promote considerable adverse events leading to poor adherence to the treatments [159,160]. Prospective and retrospective studies support the body weight-reducing effects of some of these drugs but the evidence still has considerable gaps [161]. Only a limited number of drugs (e.g., orlistat, naltrexone-bupropion, liraglutide, phentermine-topiramate, semaglutide) are currently approved by the Food and Drug Administration as an alternative therapy to promote weight loss in human adults [162]. We collected and analysed several human studies where the participants were treated with some of these anti-obesity drugs, and changes in body weight and in the levels of TNF-α were reported [29,163,164,165,166,167,168,169,170,171,172,173,174,175,176,177,178,179] (Appendix A). Our main aim was to evaluate the relevance and significance of these changes and whether there was a clear association between body weight and TNF-α in response to drug treatment. The studies were primarily conducted in obese/overweight adults of both sexes (mixed) which, in some cases, were also reported to be associated with other disorders such as T2D, hypertension, dyslipidaemia, etc. Some of the investigated trials were carried out using different chemical drugs (orlistat, L-carnitine, sibutramine, diacerein, hydroxychloroquine (HCQ), troglitazone, rosiglitazone, rimonabant, lacidipine, candesartan, metformin (MET) [29,163,164,165,166,167,168,169,170,171,172,173,174,175], and various other studies tested the effects of the protein/peptide-like molecules etanercept, exenatide, and liraglutide [176,177,178,179]. The drug intervention period varied from ≈60 d up to more than a year, and the comparison was generally done against lifestyle modifications (i.e., LCD) and/or against a widely used reference drug (i.e., MET). Most TNF-α levels were analysed in serum or plasma using the ELISA test. The TNF-α baseline values were again highly variable and oscillated between 0.9 and 132.6 pg/mL. Like in the previous sections, the results of these drug trials also showed substantial intragroup variability with most CV% values going from 9% up to >100% (mostly ranging between 30 and 50%). Except for etanercept, most of the selected studies reported a significant reduction in the circulatory levels of TNF-α both in the T and the C groups.

Orlistat, a pancreatic lipase inhibitor [180], was used in three studies at a dose of 360 mg/d to test the effects on body weight and TNF-α levels in obese participants. Of those, one study suggested a slightly better reducing effect for orlistat + LCD (T) than for LCD alone(C) (−0.5 pg/mL) but the authors did not include sufficient data to support this statement [163]. A second study indicated similar effects in the T group as well as in the C group (−4.2 pg/mL, NS) [165]; only a third study reported significant differences for the drug [164]. In this latter case, the average difference in the levels of TNF-α was ≈−12 pg/mL, and thus, the drug seemed to potentiate the effects of the diet restriction (≈2.5 fold-difference). When orlistat was combined with a second compound, i.e., L-carnitine, the effects on body weight and on TNF-α levels were not improved [166]. The average reduction in the levels of TNF-α with the combined compounds was −0.9 pg/mL (NS).

A similar limited level of evidence was found for sibutramine, a drug that stimulates satiety [181]. One study reported a reduction in body weight and TNF-α with sibutramine + LCD but a C group was not included [167]. Another study reported that two doses of sibutramine (15 mg/d and 10 mg/d) reduced body weight and TNF-α levels. In this study, the highest dose promoted a slightly higher reduction in TNF-α levels than the lower dose (−0.5 pg/mL). Nevertheless, no information on the significance of the differences between the two groups was reported, and it was not addressed whether the reduction in TNF-α was associated with body weight reduction [168]. We found one additional study that reported a significantly better reduction in body weight with sibutramine + LCD. The levels of TNF-α were decreased both in the T and the C group with the sibutramine yielding slightly better but (NS) results (−1.0 pg/mL after 1 y treatment) [169]. Studies on the effects of the other chemical drugs included in this review did not support concomitant and differential effects on body weight and TNF-α for the tested drugs. Reported differences between T and C groups oscillated between –1.1 and +0.8 pg/mL and were (NS) [29,170,171,173]. Other studies did not include sufficient data supporting TNF-α reduction [172,174]. The study by Amoani et al. [175] supported a higher and (S) decrease of the cytokine (−65 pg/mL) with a higher dose of the drug MET but this was not a proper RCT but a cross-sectional study. Regarding the protein/peptide-like drugs, etacernept, a human recombinant TNF-α inhibitor [182], was reported to be associated with an increase in the levels of TNF-α and there were (NS) differences vs. the C group changes or association with body weight [176]. Drugs of the glucagon-like peptide 1 type such as exenatide and liraglutide [177,178,179] did not show (S) and/or associated reductions in body weight and TNF-α levels. For example, treatment with exenatide + MET (T) reduced the levels of TNF-α by ≈−6.0 pg/mL vs. acarbose + MET (C), however, this difference was (NS) [178].

Overall, the data included in this section show weak evidence of the reducing effects on body weight and TNF-α of the tested anti-obesity drugs. The reported changes in TNF-α levels (‘effect size’) were small and mostly (NS). It was not possible to infer that any drug or drug-diet combination reduced the levels of TNF-α consistently and significantly, in association with body weight loss nor that the drugs were more effective than diet restriction.

## 6. General Discussion

Healthy foods and/or bioactive constituents have been long attributed with many benefits and, in particular, with anti-obesity and anti-inflammatory effects, mostly supported by many cell and animal studies reporting the reduction in the levels (gene and/or protein expression) of related biomarkers such as the pro-inflammatory cytokine TNF-α [109,110]. Despite the increasing number of intervention trials performed in this research area, proving these properties in humans remains challenging. In multiple meta-analyses and reviews looking at the beneficial effects of dietary bioactive constituents in humans, the overall recurrent messages remain the same, i.e., ‘the heterogeneity in the trial design including differences in the sample population, analytical procedures and data assessment leads to inconsistent results and increases the difficulty in interpreting those effects’, ‘interindividual variability in the responses to diet precludes the full understanding and establishment of its efficacy and true health benefits’ [106,107,112,113,183,184,185]. In the present review, we have explored the situation in a series of human intervention studies in which changes in the levels of TNF-α were investigated in response to different dietary approaches. Our aim was to contribute to the understanding of the current state of these studies and to highlight and reinforce the procedures that must be implemented in future intervention trials looking at the modulation of this important cytokine as a potential biomarker of the inflammatory response to diet. During the completion of the present article, another review looking at the effects of phytochemicals and herbal bio-active compounds on TNF-α in overweight and obese individuals was released [183]. The authors also collected and examined a number of clinical trials reporting changes in TNF-α in response to different foods and products containing mixed bioactive compounds. They concluded that there were some products with promising effects but they also found that the results were not uniform and that nearly 60% of the selected studies reported a lack of effect. In the present review, we have revisited some of those studies as well as others to critically examine the specific reported changes in the levels of TNF-α in response to dietary treatment in overweight and/or obese participants. A summary of the different intervention studies examined is presented in Table 2, where we recapitulate the data and assess the variability (CV%) of the TNF-α levels as well as the reported absolute effects (size, direction and significance). We also revise some of the main features of the trials (group sample size, source and doses of test product, intervention duration, analytical method) and the reported findings on body weight modulation for further discussion. Also, we explore the influence of a range of SNPs on the levels and variability of TNF-α.

Overall, the results of our analysis show that: (i) many of the estimated intra-group CV% values were well above 30% and, in some cases, reached values ≥100%; (ii) the circulating levels of TNF-α were highly variable and ranged from less than 1.0 pg/mL up to some examples of several hundred pg/mL; (iii) the reported changes in this cytokine following intervention with the different dietary approaches (effect) included both statistically (S) and (NS) decreases or increases of the protein with independence of the size effect, which ranged from −0.05 pg/mL up to +21.3 pg/mL. Within this intricate scenario of high intra-group and between-studies variability it was not possible to: (1) propose clear limits for the circulating levels of TNF-α as a biomarker to differentiate between lean, overweight, and obese individuals, (2) discern distinct down-regulatory effects of the levels of this cytokine by any of the different dietary approaches investigated (PUFAs, Vitamin-D, mixed (micro)nutrients, (poly)phenols or other phytochemicals), and (3) perceive what an effect-size (response) of this cytokine to dietary changes might be. In addition, we found that in most of the revised trials there was not a body weight response (body weight reduction) following intervention, nor a clear association between the body weight and the TNF-α responses. In comparison with the dietary approaches, the studies revised here looking at the effects of different anti-obesity chemical drugs and proteins showed similarities in terms of trials design and variability of the results. In addition, they showed a lack of clear response of the cytokine and of an association with body weight reduction, giving consistent evidence of the difficulty in understanding TNF-α as a responsive biomarker. Nutritional and pharmacological research should go hand-in-hand to further understand variability in the response to treatment, and to move towards more personalized anti-obesity and anti-inflammatory treatments. In the next paragraphs, we discuss some of the factors that contribute to the observed variability, and propose specific recommendations to boost the quality of future intervention trials looking at TNF-α as a biomarker of anti-inflammatory response in relationship with body weight management.

A first critical issue is ‘sample size’. It has been repeatedly indicated that ‘given the variability of the results and the generally small size effects observed in dietary interventions with bioactive constituents, larger clinical trials are needed’ [106,107,112,113,183,184,185]. Indeed, it is widely acknowledged that larger group sizes will improve the validity of the results. However, it is unclear how large they need to be to investigate changes in the levels of TNF-α? Taking into consideration the results of our revision and assuming a large overall intra-group variability CV% > 100% (i.e., 150%) and a medium size effect (*d* = 0.5), we estimated that the sample size should be ≈35 subjects per group (*p*-value < 0.05, 80% power) [186,187]. For a small size effect (*d* = 0.2), the sample size would increase up to 220 participants per arm. With a few exceptions, most of the studies investigated here were carried out in groups of less than 30 participants and, in some cases, of even less than 10 participants. Despite the well-known difficulties of recruiting and maintaining a large number of participants for this kind of intervention study, we recommend that future intervention dietary studies looking at the regulation of the levels of TNF-α in overweight/obese people must make an effort to increase the number of participants and to incorporate at least a minimum of 35 subjects in the T and C groups. Another highly variable feature regarding intervention trials looking at the beneficial effects of dietary bioactive (micro)nutrients and (or) phytochemicals is that related to the ‘duration of intervention’. It has been indicated that the greatest number of trials investigating the effects of mixed (poly)phenols have been designed as ‘chronic’ to look at the long term effects of these products [185]. However, the duration of these ‘chronic’ studies is also very variable. The reasons that lead the investigators to choose different extension periods may be various (e.g., resources availability; having a limited period of time to reduce possible toxicity, side effects or the rate of participants quitting the study; the effect might be observed quite soon; or, just the opposite, the effects might be seen in the long term, etc.). We found that the time period in the studies investigated here ranged from a minimum of 20 d up to ≈1 y, making the comparison between studies difficult. Although it is not a trivial issue to decide which intervention period is appropriate, we suggest that normalizing the intervention period to, for example, an average intermediate time of ≈6 months may provide a good opportunity to detect effects on TNF-α levels as well as to facilitate a later comparative revision or meta-analysis of those studies under review.

Intervention studies looking at the health benefits of the consumption of dietary bioactive constituents have been most commonly conducted when using as the ‘test products’ readily available mixtures of compounds in the form of enriched extracts, oils, foods, beverages. In the particular case of (poly)phenols, the greatest numbers of studies have been performed using (poly)phenol-rich foods or (poly)phenol-rich extracts, mainly those from berries, grapes, cocoa, soy, olive oil, pomegranate, flaxseed, nuts, tea or red wine [185]. The intervention studies gathered in this review looking at the response of circulating TNF-α to intervention with bioactive products have also been mostly conducted using heterogeneous mixtures of compounds (PUFAs, vitamins, minerals, proteins, phytochemicals, (poly)phenols) in different forms and doses, all of which increase the variability between studies as well as the difficulty in ascribing a particular result (effect) to a particular compound or type of compounds. This is additionally aggravated by the need to compare the T groups to appropriate C groups. Many studies report this C group as the one taking a PLA which is not always fully characterized and, sometimes, is even a different type of product, e.g., PUFAs compared to sunflower oil [63], corn starch [65] or corn oil [66]. The comparison between a similar mixture of bioactive compounds that only differentiates in the presence or concentration of the specific test compound (s) is a better control protocol that has been reported in some cases [23,118,119], but it is not an approach that can be easily and frequently implemented. Further, in comparison with drug testing (typically performed with chemically synthesized single pure molecules), not many individual pure natural bioactive compounds have been validated in humans for their effects against obesity and inflammatory conditions. For example, resveratrol is a natural bioactive (poly)phenol widely investigated for multiple health benefits including weight loss in obese individuals as well as for its anti-inflammatory properties. In a very recent report, a number of studies looking at the effects of resveratrol in obesity were critically assessed, with the conclusion reached that the results were still inconsistent due to the study design heterogeneity including major differences in many factors (sex, age, BMI, health conditions, doses, source, intervention period, etc.) [184], some of which are also further discussed and reinforced here. The authors already proposed the need to establish standardized guidelines to improve the depositing of relevant information in future trials and best practices for selecting studies that will be better used in future metanalyses. We herein reinforce the need to improve future intervention trials looking at the effects of the intake of bioactive foods and derived products on TNF-α and, for this purpose, we also propose further standardization of the study design. Along these lines, it may be worth focusing first on some individual pure test bioactive compounds such as the well-known and widely investigated resveratrol, curcumin, quercetin, or (epi)catechin, and try to definitively show evidence of their anti-inflammatory and anti-obesity effects in a specific subgroup of individuals, i.e., ‘responders’. The next step would be to replicate those results and to prove whether the compound retains its properties within an extract or a food.

It has also been recently shown that the results of the analyses of several cytokines, including TNF-α, were different across various different assays tested (singleplex, multiplex, ELISA). The authors also showed that the enzyme-linked immunosorbent ELISA assay displayed high precision and sensitivity. However, there was not a good correlation between the serum and plasma samples analysed using this method [188]. In the studies investigated in our review, the most common analytical procedure used to detect and quantify the circulating levels of TNF-α was an ELISA immunoassay (from different commercial providers) although a few other methods were also applied (multiplex, automated immuno-assay, chemiluminescence-immunoassay). The analyses were performed indistinctly in plasma or serum samples. Since the variability in the performance across different analytical platforms or in different type of samples can also greatly contribute to a lack of agreement in the reported levels of the inflammatory biomarkers, it remains essential that further investigations in this area and, in particular, in the case of TNF-α, are carried out to provide investigators with a most suitable and unique reference analytical protocol to minimise differences across studies.

The baseline characterization of the individuals taking part in a particular trial, so that at the point of departure the features of the participants are as homogeneous as possible, is an essential issue not yet fully resolved in studies looking at the beneficial properties of dietary bioactive compounds. It is still rather common to find studies carried out in mixed sample populations in terms of sex, age, body weight and/or disease status. However, each of these features constitute a well-recognised factor contributing to variability. Detailed phenotyping of trial participants for as many as possible of those potentially modulating factors has been already recommended [189]. In our revision, we focused on dietary intervention trials that investigated the potential reduction in body weight and of the levels of TNF-α; thus, most of the studies were conducted in individuals with body weight excess. Nevertheless, we found that the participants were often a mixed group of overweight and/or obese individuals and that the health status was not always clearly established or that, in some cases, the participants were additionally characterised by other metabolic or inflammatory disorders (i.e., T2D, MetS, IR, hypertension, etc.). The presence of these and other disorders can clearly contribute to the large variability in the levels of TNF-α and in the variability of the changes in this cytokine in response to the dietary intervention. In a recent meta-analysis, the values of TNF-α have been reported to vary from ≈0.4 to ≈300 pg/mL in healthy adults and from ≈0.1 to nearly 1000 pg/mL in adults suffering from Obstructive Sleep Apnea Syndrome (OSAS). These authors also reported an interaction between the disease and the BMI of the patients [190]. The data collected here regarding the different TNF-α levels between lean, overweight and obese individuals corroborate a very broad range of values and support the evidence of a large variability in the levels of this cytokine and of the difficulty in establishing clear differences between different phenotypes. The message of a thorough characterization and definition of the sample population to be investigated must be reinforced and pursued in future studies. This is particularly important when looking at individuals with body weight excess, where a much-improved definition and focus on a particular group or sub-group must be stated and supported by stricter ranges of BMI and of additional cardiometabolic health criteria. Additionally, and not always clearly specified in all the studies, the consumption of other chemicals and drugs must be clearly stated. This is especially important when looking at the differences in the levels of TNF-α between healthy subjects and overweigh/obese individuals with associated pathologies.

The presence of different SNPs in the regions involved in the regulation of the transcription of a gene might affect gene expression and protein levels, and eventually influence the circulating levels of such protein and constitute a potential associated risk to develop a disease [191]. In the case of TNF-α, many different SNPs, and in particular the rs1800629, have been long investigated for their association with the susceptibility to develop different diseases, including infections, cancer, inflammatory diseases, obesity and cardiometabolic disorders, with, however, varied and inconclusive outcomes [138]. On the other hand, there are not many studies investigating the circulating levels of the protein across the different genotypes for each polymorphism. In our review, we have gathered a number of those studies reporting the influence of different SNPs on the levels of circulating TNF-α. Overall, we corroborated a very broad range of values reported for this cytokine. Also, the effects of most of the investigated SNPs appear to be, in general, small, (NS) and highly variable. Only a few of the rare variants were associated with some (S) increases or decreases of the cytokine in specific subpopulations. Although this revision compiles only a limited number of studies and variants, it illustrates the complexity and difficulty of deciphering the overall effects of our genetic make-up on the levels of TNF-α as well as the variability attributable to this factor.

Even more complex and difficult is demonstrating the influence of genetic variants on the response to dietary intervention. Studies looking at the interaction between genotype and the intake of dietary bioactive (micro)nutrients on TNF-α are scarce and heterogeneous. In the study conducted by Curti et al. [192], the authors investigated the effect of lifestyle and dietary advice on the cytokine across the genotypes for two SNPs (TNF-α −308 G/A and IL6 −174 G/C). They reported downregulatory effects of the intervention on the cytokine but no differences between the reference and the rare variants. Similarly, de Luis et al. [139] found no differences in the levels of TNF-α across the TNF-α −308 G/A genotypes, following intervention with a diet rich in MUFAs or with a diet richer in PUFAs [69]. In a more recent intervention study with mixed vegetables containing two doses of folate, the authors reported a larger and more significant reduction in the levels of TNF-α (−5.12 pg/mL, *p* = 0.0004) only with the highest dose in the subgroup of overweight/obese women with the TT genotype for the *MTHFR* C677T polymorphism [145]. These results suggest some influence of the genotype on the response of TNF-α to the intake of folate and exemplifies a simple approach that tries to elucidate the interaction between one SNP and the dietary intake of a particular test product. Nevertheless, the situation is a lot more intricate, with so many SNPs potentially affecting the levels of TNF-α. A different and more complex approach is one that examines the effects of multiple SNPs. In a recent review [193], the interaction between 91 SNPs and the intake of fat, carbohydrates and protein on body weight loss concluded that the majority of those interactions were (NS) and thus the evidence was inconclusive. These results show once again the difficulty in finding and demonstrating the interaction between genetic variants and diet. Research on the combined effects of different SNPs on TNF-α levels and of the response of this important cytokine to intervention with dietary foods or products with anti-obesity and anti-inflammatory bioactivity is still in its infancy and requires further investigation. Some of the strategies that need to be implemented to advance the understanding of the effects of genetic variants in the human responses to dietary intervention (nutrigenetics) have been already highlighted [113]. With a combination of in vitro and animal studies, as well as molecular and genomics approaches, we need to identify the proteins and genes that are involved in the mechanisms of response as well as the list of SNPs that can affect the expression and regulation of those genes. Next, the effects of all these candidate SNPs on the levels of TNF-α in response to diet need to be evaluated in well-characterised populations carrying the different variants and in sufficiently powered trials.

One final and additional point that we would like to reinforce here is the need for and importance of increasing the quality of the data presentation [112]. While reviewing all the articles included here, we came across very different ways of data presentation with, often, important missing information. In the specific case of TNF-α, we propose that reporting the results should be normalized to the same most common units (pg/mL) and the inclusion of all the results from the intervention (control, treatment, pre-, post-, effect size, mean ± SD, CV%, *p*-values for all comparison). Requesting common protocols of reporting results will importantly contribute to the improvement of future comparative studies and the understanding of the overall outcomes.

## 7. Conclusions

TNF-α has been widely investigated in preclinical studies looking at the metabolic and anti-inflammatory benefits of a wide range of foods and bioactive food constituents yet its validation in human trials remains unsolved. A large variability in the circulating levels of the protein as well as in the response of this important cytokine to dietary intervention is partly attributable to a range of factors related with the design of the studies but also with human intrinsic causes. One such factor is the presence of a complex mixture of genetic variants that may positively and negatively influence the production of TNF-α. Improving and normalizing the study design and incorporating the genotyping of an increasing number of genetic variants into the intervention trials will contribute to understanding the large interindividual variability of the cytokine as well as to improving the comparative analyses of multiple results from different studies, increasing the evidence and validity of TNF-α as a biomarker of response to diet.

## Figures and Tables

**Figure 1 foods-11-02524-f001:**
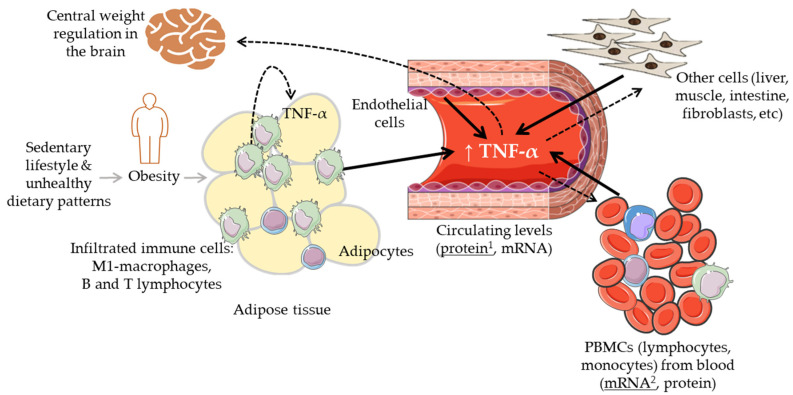
Graphic summary of the biological regulatory scenario of the levels of TNF-α in the context of body weight alterations. Solid black arrows indicate different cells and tissues that can contribute to the circulating levels of TNF-α; dashed arrows indicate different tissues where TNF-α can have a regulatory effect. ^1^: TNF-α protein levels most commonly measured in blood (serum, plasma); ^2^: *TNF-α* mRNA levels most commonly measured in peripheral blood mononuclear cells (PBMCs).

**Figure 2 foods-11-02524-f002:**
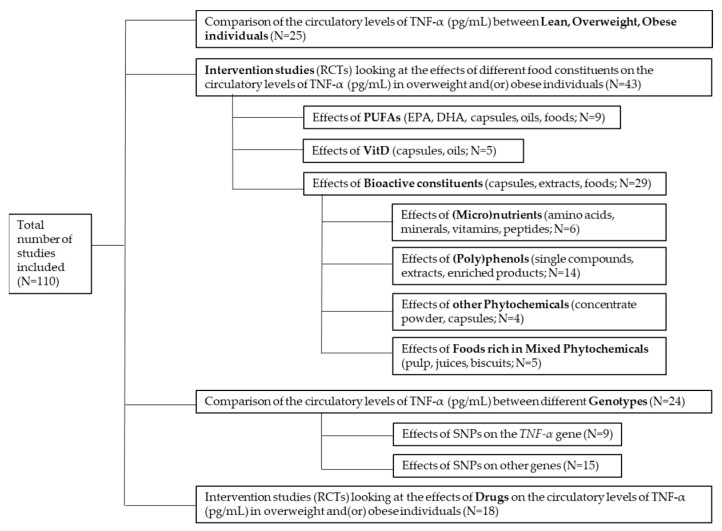
Diagram summarising the studies revised in this Review (N = number of studies included).

**Table 1 foods-11-02524-t001:** Summary of the data on intragroup variability and levels of circulatory TNF-α in lean, overweight, obese, and very obese individuals as reported in the different human studies examined in this review.

Phenotype	CV% ^1^ (Range)	Most Commonly CV% Values (Range)	TNF-α Levels (Range)	Most Commonly Reported TNF-α Levels (Range)
Lean	(12%, ≥100%)	(≈30%, 50%)	(0.09, 82.3 pg/mL)	(≈2.0, 6.0 pg/mL)
Overweight	(37%, >100%)	(≈35%, 50%)	(0.09, 30.0 pg/mL)	(≈3.0, 6.5 pg/mL)
Obese	(11%, >100%)	(≈30%, 90%)	(0.11, 294.0 pg/mL)	(≈1.0, 10.0 pg/mL)
Very obese ^2^	(21%, >100%)	(≈30%, 90%)	(1.3, 713.0 pg/mL)	(≈5.0, 10.0 pg/mL)

^1^: CV% = (Standard Deviation/Mean) × 100; ^2^: This group includes participants classified as very obese, class II and III obesity, and morbidly obese.

**Table 2 foods-11-02524-t002:** Summary of the results from the human intervention studies gathered in this review looking at the effects of different dietary products and several anti-obesity drugs on the circulatory levels of TNF-α in overweight/obese participants in relationship with body weight management.

	Study Characteristics				TNF-α Levels, Variability, Change			
Dietary approach	Number of RCTs (*N*: range of size per arm in the RCTs)	Doses (range) Duration (range)	Population phenotype	Samples Method	Baseline levels (range)	Intragroup variability (CV% range)	Effect description	
							Average difference T–C (pg/mL range) Significance (S/NS)	Overall reported body weight message
(Micro)nutrients								
ω-3 PUFAs (EPA, DHA; oils capsules)	5 RCTs (*11*, *49*)	(500, 4000 mg/d) (30, 180 d)	Overweight/Obese/Obese (IR, T2D)	Serum, plasma ELISA	(1.09, 25.8 pg/mL)	(13%, 59%)	−0.5 (S) (−0.02, +10.9) (NS)	(NS) effects on body weight.
PUFAs (foods)	4 RCTs (7, *45*)	(3, 50 g/d) (56, 84 d)	Overweight/Obese/Obese (glucose intolerant)	Serum, plasma ELISA, Milliplex	(0.6, 24.0 pg/mL)	(5%, 86%)	−6.3 (S) (−0.2, +0.7) (NS)	(NS) effects on body weight.
Vitamins (e.g., Vitamin D)	5 RCTs (*10*, *83*)	(0.075, 0.1 mg/d) (56, 365 d)	Overweight/Obese/Obese (IR)	Serum, ELISA, Automated immuno-assay	(1.7, 39.1 pg/mL)	(7%, >100%)	−0.6 (S) (−3.5, +5.9) (NS)	(NS) effects on body weight.
Other mixed (micro)nutrients (amino acids, peptides, minerals, complex polysaccharides, etc.)	6 RCTs (*10*, *35*)	(30 mg/d, 9 g/d) (28, 90 d)	Overweight/Obese/Obese (MetS)	Serum, plasma ELISA	(0.22, 411 pg/mL)	(5%, 47%)	(−56, +0.61) (NS)	(S) effects on body weight only with black soy peptide. (S) WC reduction only with yeast β-glucan
Phytochemicals								
Mix extracts, powders rich in (poly)phenols	15 RCTs (*8*, *93*)	(15, 600 mg/d) (21, 360 d)	Overweight/Obese/Overweight/obese (cardiac, CAD, T2D, MetS, hypertension, knee OA)	Serum, plasma ELISA, Chemiluminesceimmunoassay	(0.13, 43.0 pg/mL)	(4%, >100%)	(−0.05, −11.9) (S) (−2.6, +2.6) (NS)	(S) effects on body weight only with a grape seed extract (85% (poly)phenols).
Mix extracts, powders rich in other phytochemicals	4 RCTs (*16*, *92*)	Variable quantities (mg/d) (40, 112 d)	Mixed Overweight/Obese/Overweight/obese (knee OA)	Serum, plasma ELISA	(1.04, 62.8 pg/mL)	(4%, >100%)	(−0.07, −0.98) (S) (−0.21, +14.0) (NS)	(NS) effects on body weight.
Foods/beverages containing mixed phytochemicals	5 RCTs (*10*, *53*)	(5, 45 g/d; 250, 330 mL/d) (20, 84 d)	Overweight/obese Overweight/obese (hypertension)	Serum, plasma ELISA	(2.9, 73.2 pg/mL)	(12%, 100%)	(−2.0, −8.3) (S) (−0.21, +21.3) (NS)	(NS) effects on body weight.
Anti-obesity drugs								
Chemical drugs	11 RCTs (*7*, *190*)	(2, 2000 mg/d) (56, 570 d)	Obese Obese (T2D, hypertension, dyslipidaemia)Overweight/obese (IR, T2D, PCOS, hypertension)	Serum, plasma ELISA, Bioplex, EIA	(2.94, 132,6 pg/mL)	(9%, >100%)	(−12.1, −65.0) (S) (−4.2, +0.8) (NS)	Unclear effects on body weight and no association with TNF-α.
Protein/peptide drugs	4 RCTs (*12*, *54*)	(µg to mg/d) (s.c.i.) (84, 180 d)	Overweight/Obese/Overweight/obese (CAD, T2D)	Serum, plasma ELISA	(0.9, 34.6 pg/mL)	(36%, >100%)	ND (−6.0) (NS)	Unclear effects on body weight and no association with TNF-α.

CV% = (Standard Deviation/Mean) × 100; CAD: coronary artery diseases; IR: insulin resistance; MetS: Metabolic Syndrome; PCOS: Polycystic Ovary Syndrome; T2D: Type 2 Diabetes mellitus; OA: Osteoarthritis.

## Data Availability

The data presented in this review are available in the article and in the Appendix A.

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
