# Peer review of "An Exploratory Critical Review on TNF-α as a Potential Inflammatory Biomarker Responsive to Dietary Intervention with Bioactive Foods and Derived Products"

_foods, 2022, doi:10.3390/foods11162524_

Round 1
Reviewer 1 Report
The title of this article is “An exploratory critical review on TNF-α as a potential inflammatory biomarker responsive to dietary intervention with bioactive foods and derived products”. This is an interesting topic, and it is an area that needs our attention. However, there are still some areas of the article that need to be revised.
1. In the "2. TNF-α levels in lean, overweight and obese individuals" section of the article, the authors need to introduce more examples to illustrate the differences in TNF-α levels between obese and healthy individuals.
2. The article talks about the effect of vitamin D on TNF-α levels, which is an interesting part. The authors could have started with common foods rich in vitamin D in people's lives and cited more examples to prove the beneficial effects of vitamin D.
3. Polyphenols in food have been a popular topic of research and their effects on health have been somewhat confirmed. The authors could degrade the regulatory effect of polyphenols on TNF-α levels in more ways, such as polyphenols affecting intestinal flora to improve body metabolism.
4. In the article "4. The influence of genetic variants on TNF-α levels", the authors need to explore the content of this section in more depth and give more of their own opinions and perspectives for the future.
5. Authors are requested to carefully check the format of the references used in the article to ensure that the references are in the required format.
Author Response
Reviews
Manuscript ID: foods-1864095
Type of manuscript: Review
Title: An exploratory critical review on TNF-α as a potential inflammatory
biomarker responsive to dietary intervention with bioactive foods and derived
products.
Authors: Stefano Quarta, Marika Massaro, Maria Annunziata Carluccio, Nadia
Calabriso, Laura Bravo, Beatriz Sarria, María-Teresa García-Conesa *
Reviewer 1
Comments and Suggestions for Authors
The title of this article is “An exploratory critical review on TNF-α as a potential inflammatory biomarker responsive to dietary intervention with bioactive foods and derived products”. This is an interesting topic, and it is an area that needs our attention. However, there are still some areas of the article that need to be revised.
We thank the Reviewer for taking the time and efforts to read and revise our manuscript. Please, see below and within the manuscript (marked in red) the changes carried out following her/his suggestions.
- In the "2. TNF-α levels in lean, overweight and obese individuals" section of the article, the authors need to introduce more examples to illustrate the differences in TNF-α levels between obese and healthy individuals.
The results of all the studies evaluated in this section and the differences between obese and lean individuals for each study are specified in the supplementary Table S1. This has been clarified with a sentence in page 4, lines 144-145. Nevertheless, and as suggested by the reviewer, we have also indicated in the text some of those results as examples of the variability of the levels and changes in TNF-α between lean and obese participants (page 5, lines 172-188).
- The article talks about the effect of vitamin D on TNF-α levels, which is an interesting part. The authors could have started with common foods rich in vitamin D in people's lives and cited more examples to prove the beneficial effects of vitamin D.
Following the Reviewer’s recommendation, we have indicated more clearly some of the main foods and products containing VitD, and updated the general knowledge on the role and effects of VitD in various diseases (page 7, lines 314-323; page 8, lines 325-328).
- Polyphenols in food have been a popular topic of research and their effects on health have been somewhat confirmed. The authors could degrade the regulatory effect of polyphenols on TNF-α levels in more ways, such as polyphenols affecting intestinal flora to improve body metabolism.
We thank the Reviewer for this comment. We have now included the information on the role of (poly)phenols on inflammatory and obesity, highlighting the importance of understanding the double interaction between these compounds and the gut microflora as a major factor influencing the regulatory effects of these compounds (page 9, lines 427-432).
- In the article "4. The influence of genetic variants on TNF-α levels", the authors need to explore the content of this section in more depth and give more of their own opinions and perspectives for the future.
We have now included some of the strategies that have been proposed in order to advance in the understanding of the effects of genetic variation in the response to foods (page 17, lines 847-855).
- Authors are requested to carefully check the format of the references used in the article to ensure that the references are in the required format.
We thank the Reviewer for this comment. There were indeed many mistakes in the numbers of the references that have now all been corrected. Three new references have also been included. The list of references in the Supplementary material has been deleted and now the references of these tables are all in the main list of references of the manuscript.
Reviewer 2 Report
Research about the role of TNF- α in obesity is still a subject of active research work. This narrative review aims to summarize the recent data regarding TNF-α variability levels and the responses to food intervention reported in Literature. It is a well-conducted study and a well-written manuscript. There is a few comments to make the manuscript better:
- In section 2, a PRISMA diagram should highlight the inclusion and exclusion criteria for revised studies carried out by the authors (please see lines 126-128).
- Are the "very obese" group include severe obesity and morbid obesity? (Please see table 1)
- To estimate the coefficient of variation (CV%) for each of the different subgroups, has been used a soft program? Please clarify this aspect.
- In lines 106 and 107, please add references regarding the data presented.
Author Response
Reviews
Manuscript ID: foods-1864095
Type of manuscript: Review
Title: An exploratory critical review on TNF-α as a potential inflammatory
biomarker responsive to dietary intervention with bioactive foods and derived
products.
Authors: Stefano Quarta, Marika Massaro, Maria Annunziata Carluccio, Nadia
Calabriso, Laura Bravo, Beatriz Sarria, María-Teresa García-Conesa *
Reviewer 2
Comments and Suggestions for Authors
Research about the role of TNF- α in obesity is still a subject of active research work. This narrative review aims to summarize the recent data regarding TNF-α variability levels and the responses to food intervention reported in Literature. It is a well-conducted study and a well-written manuscript. There are a few comments to make the manuscript better:
We thank the Reviewer for taking the time and efforts to read and revise our manuscript. Please, see below and within the manuscript (marked in red) the changes carried out following his/her suggestions.
- In section 2, a PRISMA diagram should highlight the inclusion and exclusion criteria for revised studies carried out by the authors (please see lines 126-128).
This article is an exploratory review rather than a systematic review and thus, it was not strictly conducted following the PRISMA guidelines. When we started looking for articles looking at TNF-α responses to diet, we realized there was a very high heterogeneity of studies and results and, at that point, we decided that it would be more helpful to conduct a critical revision of all the studies taking into consideration the variability of those results. Nevertheless, and following the suggestion of the Reviewer, we have included a diagram (Figure 2) with a summary of all the studies examined in this review. We hope that this can help the readers to more clearly follow the sequence of the article.
- Are the "very obese" group include severe obesity and morbid obesity? (Please see table 1)
The classification of the level of obesity of the participants from the different studies examined here is included in Supplementary Table S1. In most cases, the authors referred to the participants as “obese” but we found one study that included participants with “morbid obesity” (ref. 17 in Supplementary Table S1) and, in several other cases, some of the participants were classified as “very obese” or with different classes (class I, II and III) of obesity (refs. 2, 3, 16, 17, 20, 25 in Supplementary Table S1). In Table 1, we grouped as “very obese” all those classified as very obese, class II and III, and morbidly obese to see whether there was any difference from the “obese” category. We have added a note in the legend to clarify this point.
- To estimate the coefficient of variation (CV%) for each of the different subgroups, has been used a soft program? Please clarify this aspect.
The CV% for each subgroup was calculated using the standard formula as follows: Coefficient of Variation (CV) = (Standard Deviation/Mean) × 100. This sentence has been added for clarification within the text (page 4, lines 139-140) and as part of the legend for Table 1, Table 2 and for each Supplementary Table.
- In lines 106 and 107, please add references regarding the data presented.
These lines were a general sentence to introduce the objective of the article and did not need references. Nevertheless, we have modified the sentence and have tried to clarify this issue with the addition of the diagram in Figure 2.